# Noninvasive Tracking of Hematopoietic Stem Cells in a Bone Marrow Transplant Model

**DOI:** 10.3390/cells9040939

**Published:** 2020-04-10

**Authors:** Fernando A. Oliveira, Mariana P. Nucci, Igor S. Filgueiras, João M. Ferreira, Leopoldo P. Nucci, Javier B. Mamani, Fernando Alvieri, Lucas E. B. Souza, Gabriel N. A. Rego, Andrea T. Kondo, Nelson Hamerschlak, Lionel F. Gamarra

**Affiliations:** 1Hospital Israelita Albert Einstein, São Paulo 05652-900, Brazil; fernando.anselmo@einstein.br (F.A.O.); igor.filgueiras@usp.br (I.S.F.); joaomatiasferreirav@gmail.com (J.M.F.); javierbm@einstein.br (J.B.M.); fernando.alvieri@einstein.br (F.A.); gabriel.nery@einstein.br (G.N.A.R.); andrea.kondo@einstein.br (A.T.K.); hamer@einstein.br (N.H.); 2LIM44—Hospital das Clinicas HCFMUSP, Faculdade de Medicina, Universidade de São Paulo, São Paulo 01246-903, Brazil; mariana.nucci@hc.fm.usp.br; 3Centro Universitário do Planalto Central, Brasília DF 72445-020, Brazil; leopoldo.nucci@gmail.com; 4Faculdade de Medicina de Ribeirão Preto, Universidade de São Paulo, Ribeirão Preto SP 14049-900, Brazil; lucasebsouza@usp.br

**Keywords:** hematopoietic stem cell, nanoparticle, homing, tracking, near-infrared fluorescence image, magnetic resonance image, bioluminescence, molecular imaging, noninvasive imaging

## Abstract

The hematopoietic stem cell engraftment depends on adequate cell numbers, their homing, and the subsequent short and long-term engraftment of these cells in the niche. We performed a systematic review of the methods employed to track hematopoietic reconstitution using molecular imaging. We searched articles indexed, published prior to January 2020, in PubMed, Cochrane, and Scopus with the following keyword sequences: (Hematopoietic Stem Cell OR Hematopoietic Progenitor Cell) AND (Tracking OR Homing) AND (Transplantation). Of 2191 articles identified, only 21 articles were included in this review, after screening and eligibility assessment. The cell source was in the majority of bone marrow from mice (43%), followed by the umbilical cord from humans (33%). The labeling agent had the follow distribution between the selected studies: 14% nanoparticle, 29% radioisotope, 19% fluorophore, 19% luciferase, and 19% animal transgenic. The type of graft used in the studies was 57% allogeneic, 38% xenogeneic, and 5% autologous, being the HSC receptor: 57% mice, 9% rat, 19% fish, 5% for dog, porcine and salamander. The imaging technique used in the HSC tracking had the following distribution between studies: Positron emission tomography/single-photon emission computed tomography 29%, bioluminescence 33%, fluorescence 19%, magnetic resonance imaging 14%, and near-infrared fluorescence imaging 5%. The efficiency of the graft was evaluated in 61% of the selected studies, and before one month of implantation, the cell renewal was very low (less than 20%), but after three months, the efficiency was more than 50%, mainly in the allogeneic graft. In conclusion, our review showed an increase in using noninvasive imaging techniques in HSC tracking using the bone marrow transplant model. However, successful transplantation depends on the formation of engraftment, and the functionality of cells after the graft, aspects that are poorly explored and that have high relevance for clinical analysis.

## 1. Introduction

Studies from the early 1950s established that total body irradiation in animal models causes death from hemorrhage and infection, indicating that the hematopoietic system is primarily affected [1]. However, it was also shown that transplantation of genetically identical (i.e., syngeneic) bone marrow cells rescues these animals from death induced by irradiation [1].

Later on, Edward Donnal Thomas and colleagues pioneered the application of the results from these early animal studies for the treatment of leukemia in humans. The approach used here was to kill leukemic cells by high-dose irradiation, followed by restoration of the hematopoietic system with bone marrow transplantation [2]. These early findings provided the rationale for using hematopoietic stem cell transplantation (HSCT) as the first stem cell-based therapy for the treatment of a wide plethora of hematopoietic disorders.

According to a comprehensive report from the Worldwide Network for Bone Marrow Transplantation (WBMT), by the end of 2012, more than one million patients had undergone HSCT [3]. The vast majority of HSCT transplantation procedures were used to treat malignant disorders (87%), most of them leukemias (72%), followed by lymphoproliferative disorders (14.7%) and solid tumors (0.6%) [3]. It is noteworthy that HSCT also cures several genetic diseases, such as severe combined immunodeficiency, Wiskott–Aldrich syndrome, thalassemia, and sickle-cell anemia [4].

The dissemination of HSCT as a therapeutic modality is closely linked to the identification and typing of the major histocompatibility complex (also termed human leukocyte antigens (HLA)) in the early 1960s. As a consequence of these discoveries, allogeneic transplantation of HSCs between HLA-matched individuals became feasible. Indeed, almost half of HSCT procedures are allogeneic according to the latest global survey [3].

Allogeneic HSCT comes with the risk of developing a serious immune reaction termed graft versus host disease (GVHD), in which alloreactive donor T cells attack the recipient’s tissues [5]. GVHD is the primary immune barrier to allogeneic HSCT efficacy and is the second cause of death in patients that undergo this procedure, falling behind only mortality caused by the primary disease [6]. For autologous HSCT, on the other hand, the main factor limiting its efficacy is graft failure. Graft failure is a rare complication of HSCT and may be caused by several factors, such as a low dose of injected HSCs, old HSC donors, bone marrow fibrosis in the recipient, storage techniques affecting HSC integrity, and pre-HSCT treatment with chemotherapy and/or irradiation [7].

Studying the dynamics of HSC engraftment and of its progeny is of paramount importance to define the mechanistic basis of hematopoietic reconstitution and the complications of HSCT. For a long time, most of the data regarding the engraftment and expansion of HSCs in small animals were obtained from post-mortem analysis of hematopoietic organs. With the development of imaging hardware, molecular imaging, and imaging reporters, HSCs could be tagged with radioisotopes, fluorophores, contrast agents, reporter constructs, ligands, probes, or cell transduction by reporter genes that codified proteins, such as GFP and luciferase, allowing migration homing and tracking upon transplantation [8,9,10,11,12]. These advances have provided both temporal and spatial information of experimental HSCT that no other technique could provide. In addition, molecular imaging involves a set of noninvasive techniques that allow a serial analysis of the same individual, thereby significantly decreasing the number of animals used for experimentation [13,14,15].

To study the dynamics of hematopoietic reconstitution, in which cells localize deep inside hematopoietic organs and other tissues, different modalities of molecular imaging are used for HSC tracking and engraftment, such as (i) optical imaging represented by bioluminescence imaging (BLI) [9,11,16,17,18,19,20] and fluorescence imaging (FLI) [21,22,23,24,25], (ii) nuclear imaging represented by positron emission tomography (PET) [26,27,28] and single-photon emission computed tomography (SPECT) [29,30,31], and (iii) magnetic resonance imaging (MRI) [32,33,34].

BLI generally provides high sensitivity with a high signal-to-noise ratio when compared to fluorescent imaging [35]; moreover, its tissue penetrability in higher than that of fluorescence imaging [14,36], and has a high throughput for imaging of small animals [9,15], as also a wide temporal detection window (0 days to 1 year) [9], despite its use being limited to preclinical studies. In vivo BLI enables real-time monitoring of gene expression and cell fate through visual representation of the bioluminescence generated by oxidation of specific substrates by luciferase enzymes, providing a dynamic profile of engraftment and proliferation in live recipient animals [9,37]. Thus, cells genetically engineered to express luciferase can be tracked after substrate injection using highly sensitive charged-coupled device (CCD) cameras to detect light [36]. On the other hand, FLI can be obtained using fluorescent probes, which, when in the visible length, makes it is possible to detect by intravital microscopy techniques, and when the probes emit a fluorescence signal in the wavelength between 650 and 900 nm (infrared spectrum), it is possible to acquire images of deeper tissues compared to conventional fluorescence imaging [38,39]. Thus, the near-infrared fluorescence (NIRF) imaging has a temporal resolution in the order of s to min, a spatial resolution of 2 to 3 mm with a depth of penetration less than 1 cm, low sensibility, no use of ionizing radiation, and low cost. The NIRF disadvantages are the same as that of the BLI modality (low spatial resolution and low tissue penetration), and also as that of autofluorescence [40,41]. Most of the studies used this technique for early HSC engraftment evaluation in the mouse models, but mainly fishes [21,22,23,24,25].

The nuclear image modalities, PET and SPECT are extremely valuable for investigating molecular processes in vivo, which are based on the administration and detection of radioisotopes. However, the HSCT tracking time depends on the radioisotope half-life, resulting in a small temporal window in signal detection. These techniques have good sensitivity, high penetration without depth limit, and can be translated into clinical practice, but they have a low spatial resolution, require the use of ionizing radiation, and have a high cost [14,42,43].

MRI provides excellent anatomical resolution, in vivo cell tracking and graft-host with 3-dimensional information, high penetration depth, does not require the use of ionizing radiation, and is easy to translate into clinical application, although this modality is limited by low sensitivity and high instrumentation cost, as well the tracer dilution upon cell division. MRI is now emerging and rapidly expanding in all fields. In addition, it has the advantages of safety, high spatial and temporal resolution, and direct applicability to cell tracking in clinical studies. MRI specificity can be increased by the use of the contrast agents with magnetic character (paramagnetic and superparamagnetic) [44,45,46,47,48].

Molecular imaging techniques vary in terms of their characteristics of detecting the corresponding signal (specificity, sensitivity, and depth), but technological advances in the development of new multifunctional contrast agents have allowed the use of combined imaging techniques, providing more information and complementing the results obtained, thus culminating in greater clarity of the location of the cells, in addition to allowing co-registration [46,49,50].

In addition, luciferase can be coupled to nanoparticles and fluorescent proteins to create multimodality reporters or brighter probes [51]. One interesting example is the process of generating bioluminescence resonance energy transfer (BRET) upon the fusion between Renilla luciferase and the fluorescent protein Venus. The resulting structure is an auto-illuminated fluorescent probe that allows subcellular imaging at high resolution and, at the same time, displays enhanced in vivo brightness when compared to Renilla luciferase alone [52].

The potential of molecular imaging to reveal complex biological processes is virtually endless. These applications dramatically enhance the amount of information that can be easily obtained from refined animal models. Therefore, this review aimed at performing a systematic review of the methods employed to track hematopoietic reconstitution using molecular imaging. We hope that this effort will serve as a practical and comprehensive guide to help researchers to refine their in vivo models of HSCT by incorporating molecular imaging strategies.

## 2. Materials and Methods

### 2.1. Search Strategy

This systematic review follows the Preferred Reporting Items for Systematic Reviews and Meta-Analyses (PRISMA) guidelines [53]. We searched articles indexed, published prior to January 2020, in PubMed, Cochrane, and Scopus. The following selected criteria of interest, keyword sequences ((Hematopoietic Stem Cell OR Hematopoietic Progenitor Cell) AND (Tracking OR Homing) AND (Transplantation)), and boolean operators (DecS/MeSH) were used:

SCOPUS: ((TITLE-ABS-KEY (tracking) OR TITLE-ABS-KEY (homing))) AND ((TITLE-ABS-KEY (“Hematopoietic Stem Cell”) OR TITLE-ABS-KEY (“Hematopoietic Stem Cells”) OR TITLE-ABS-KEY (“Hematopoietic Cell”) OR TITLE-ABS-KEY (“Hematopoietic Cells”) OR TITLE-ABS-KEY (“Hematopoietic Progenitor Cell”) OR TITLE-ABS-KEY (“Hematopoietic Progenitor Cells”))) AND (TITLE-ABS-KEY (transplantation))

PubMed: (((((“Hematopoietic Stem Cell”[Title/Abstract]) OR “Hematopoietic Stem Cells”[Title/Abstract]) OR “Hematopoietic Progenitor Cell”[Title/Abstract]) OR “Hematopoietic Progenitor Cells”[Title/Abstract])) AND ((tracking[Title/Abstract]) OR homing[Title/Abstract])) AND transplantation[Title/Abstract]

Cochrane: “hematopoietic stem cell” in Title Abstract Keyword OR “hematopoietic progenitor cell” in Title Abstract Keyword AND tracking in Title Abstract Keyword OR homing in Title Abstract Keyword AND “transplantation” in Title Abstract Keyword—(Word variations have been searched)

### 2.2. Inclusion Criteria

Eligibility criteria were established a priori. This review included only original articles written in English, published between 2000 and 2020, that had used (i) in vivo models of bone marrow transplantations, (ii) tracking agents to analyze the hematopoietic stem cells labeled with these agents, and (iii) noninvasive techniques to allow the in vivo homing of hematopoietic stem cells in the bone marrow transplantation model.

### 2.3. Exclusion Criteria

Reasons for excluding studies were as follows: (i) reviews, (ii) clinical articles, (iii) book chapters, (iv) protocols, (v) editorials/expert opinions, (vi) letters/communications, (vii) publications in languages other than English, (viii) did not analyze stem cell homing with noninvasive techniques, (ix) indexed articles published in more than one database (duplicates), (x) only used invasive techniques to analyze hematopoietic stem cell homing.

### 2.4. Data Extraction, Data Collection, and Risk of Bias Assessment

In this review, seven of the authors (F.A.O.; M.P.N.; I.S.F.; J.B.M.; F.A.; G.N.A.R. and L.F.G.) independently and randomly selected (in pairs), revised, and evaluated the titles and abstracts of the publications identified by the search strategy in the databases cited above, and all potentially relevant publications were retrieved in full. These same reviewers evaluated the full text articles to decide whether the eligibility criteria were met. Discrepancies in study selection and data extraction between the two reviewers were discussed with a third reviewer and resolved.

F.A.O.; M.P.N.; J.M.F. and L.E.B.S searched for in vivo models of bone marrow transplantations; F.A.O.; I.S.F.; J.B.M., G.N.A.R. and L.P.N. searched for tracking agents to analyze the hematopoietic stem cells labeled with these agents; F.A.O.; M.P.N.; I.S.F.; F.A., and L.F.G. searched for noninvasive techniques to allow the in vivo homing of hematopoietic stem cells in the bone marrow transplantation model. The analysis process and table plots were carried out by a full consensus of peers, respecting the distribution above. In cases of disagreement, a third, independent author decided to add or subtract data. The final inclusion of studies into the systematic review was by agreement of all reviewers.

### 2.5. Data Analysis

All results were described and presented using the percentage distribution for all variables analyzed in the tables.

## 3. Results

### 3.1. Selection Process of the Articles Identified According to the PRISMA Guidelines

We searched publications between January 2000 and January 2020, indexed in PubMed, Scopus, and Cochrane Library, and a total of 2191 articles were identified. Of the 50 articles identified in Cochrane Library, none were included because, out of the articles, 42 were reviews, 5 were only protocols, and 3 were clinical articles. Of the 447 articles identified in Pubmed, 114 were excluded after screening (82 reviews, 24 publications before 2000, and 8 publications in other languages) and 325 articles were excluded after assessing eligibility (91 reported no data about HSC, 99 reported no data about animal models, and 135 reported no data about noninvasive imaging); thus, only 8 articles were included from this database. Of the 1694 articles identified in Scopus, after screening, 771 articles were excluded (416 reviews, 106 publications before 2000, 50 publications in other languages, and 199 duplicated in Pubmed search), and after assessing eligibility, 910 articles were excluded (287 reported no data about HSC and 623 reported no data about noninvasive imaging); thus, only 13 articles were included from this database. In total, only 21 nonduplicate full text articles were included in this review [9,11,16,17,18,19,20,21,22,23,24,25,26,27,28,29,30,31,32,33,34], as depicted in Figure 1.

### 3.2. Extraction and Isolation of Hematopoietic Stem Cells

The hematopoietic stem cell (HSC) extraction and isolation process used in the selected studies is described in detail in Table 1. Of the 21 selected studies, nine (43%) [16,18,19,20,23,25,28,32,34] studies used mice as a cell donor for bone marrow transplant (BMT), seven studies (33%) used humans as a cell donors [9,11,21,24,26,27,33], two studies (10%) used rat cells [29,30], and the remaining studies used other cell donors, such as fish [17], axolotl [22], and dogs [31], corresponding 5% for each study. Regarding the source of the cells, the majority of studies (53%) [16,18,19,20,23,25,28,29,30,31,32] used bone marrow; however, the study by Ushiki [25] used bone marrow and spleen cells, and the second main source of cells was the umbilical cord used in five studies (24%) [9,11,24,27,33], followed by peripheral blood, which was used in two studies (10%) [21,26]. Furthermore, the study by Lopes [22] used liver and spleen cells, the study by Astuti [17] collected the renal marrow cells from zebrafish, and the study by Sweeney [34] obtained cells from embryonic cells. Of eleven studies that used rodents as cell donors, the age of the animals was, on average, 8 weeks [18,20,23,25,30], ranging from 6 [18] to 10 [20] weeks, and the use of male rodents [29,30,31,32] was more prevalent than of female rodents [18], though two studies [23,28] used both genders. In the study by Lange [31], which used dogs, the animals were all male of –-3 years of age, and of the studies that used humans as cell donors, only the study by Massolo [29] reported the age range and gender of the volunteers (aged 18–40 and both genders).

The main method used for harvesting stem cells from animals in the selected studies was flushing (29%) [18,29,30,32], followed by maceration (21%) [22,23,28]. The other studies reported the methods of differentiation from embryonic cells [34] and femoral bone marrow aspiration in live animals [31]. Of the studies that used humans as cell donors, only one study [26] reported the apheresis method for harvesting stem cells.

Concerning the medium and supplementation used during the harvesting of stem cells, two studies [29,30] used Dulbecco’s modified Eagle medium (DMEM) supplemented with 10% fetal bovine serum (FBS) and heparin. Three other studies [17,22,23] used Phosphate-buffered saline (PBS), and the study by Parada-Kusz [23] supplemented the PBS with 0.5% bovine serum albumin (BSA)/FBS and 2 mM EDTA (Ethylenediaminetetraacetic acid). Furthermore, the study by Lopez [22] used 0.75× PBS supplemented with 5% BFS (APBS, axolotl PBS); the study by Lin [18] used minimum essential medium Eagle alpha modification (α-MEM) supplemented with 10% heparin; the study by Sweeny [34] used StemPro34 medium with a hematopoietic cytokine selection; and in the study by Niemeyer [33], the harvesting of stem cells was performed with citrate buffer.

For isolating mononuclear cells from the pool of cells collected, most of the selected studies (33%) [11,22,24,29,30,31,33] used the Ficoll density gradient method, although the study by Ushiki [25] used Lympholyte, and the study by Hamilton [21] used Percoll for HSC isolation. The phenotypic characterization identified the primitive lineage in all of the studies that used the isolated cells from humans as CD34 positive (+) [9,11,21,24,27,33] and CD38 negative (–) [9], as well as a subpopulation of CD34(+) and CD133 positive(+) [26]. The same primitive lineage was reported in three studies from rat-isolated cells as CD90 positive (+) [29,30], and dog-isolated cells as CD34(+) [31]. However, in the studies that used cells from mice, the phenotypic characterizations identified the primitive lineage as an early form of mouse hematopoietic stem cells, such as KSL cells [19,32], (c-kit positive (+) [23,34], Sca-1 positive (+) [34], Lin negative (–) [20]), as well as CD45(+) [23,34], CD11b(+) [23], and CD41(+) [34]. The study by Lange [31] identified the isolated cells as the primitive lineage of CD34(+). The cell lines of interest were isolated by sorting in six studies (29%) [20,23,25,26,29,33], using only Magnetic-activated cell sorting (MACS). Another three studies (14%) [22,29,30] used Fluorescence-activated cell sorting (FACS), and a further six studies (29%) [9,11,19,21,24,32] (29%) used both techniques (FACS and MACS). Only four studies [11,24,27,33] reported HSC-isolated cell purity superior to 95%, and one study [9] reported purity between 80% and 90%.

### 3.3. Lentiviral Transduction of Hematopoietic Stem Cells

One of the noninvasive techniques used for HSC tracking after bone marrow transplantation is bioluminescence imaging. This technique depends on HSCs expressing the luciferase enzyme that, added to its substrate under ideal conditions, can emit light. This technique was used in 7 out of the 21 studies [9,11,16,17,18,19,20] for HSC migration homing and tracking analysis (Table 2). Three studies [16,17,20] used genetically modified animals that expressed the luciferase enzyme, and the other studies [9,11,18,19] performed luciferase transduction in hematopoietic stem cells to express the bioluminescence signal. The transduction process was performed with different vectors; for instance, the study by Lin [18] used a lower multiplicity of infection (MOI) ranging from 0.5 to 1, and the study by Ohmori [19] used an MOI of 20. Thus, already, the cell dose ranges from 3–30 × 10^2^ [11] to 3–8 × 10^5^ per well. For improving the transduction efficiency, three of four studies used the transfection agent during the process, such as polybrene (8 µg/mL) [18,19] and retronectin (100 µg/mL) [11], and the culture medium and supplementation varied among studies, as well as the cytokines used in the transduction process. The time of incubation was mainly 24 h, with the exception of the study by Lim [18], which used 48 h, and the study by Wang [9], which used two cycles of 24 h with an interval of 8 h. Luciferase expression evaluation was performed mainly, in three studies [11,18,19], by the flow cytometry technique (FCT); however, the study by Lin [18] also used the polymerase chain reaction (PCR) and colony forming unity (CFU) techniques, and the study by Wang [9] conducted only immunohistochemistry using monoclonal antiluciferase antibody.

### 3.4. Labeling Strategies and Techniques of Hematopoietic Stem Cells with Tracers

The labeling of HSC with tracers is one way to analyze cell tracking by other noninvasive image techniques. Of the 21 selected studies in this review, in 13 studies (62%) [21,23,24,25,26,27,28,29,30,31,32,33,34], the HSCs were submitted to the labeling process using various types of tracers, such as radioisotope agents used for PET/SPECT imaging in six studies (46%) [26,27,28,29,30,31], fluorophore agents used in FLI imaging in four studies (31%) [21,23,24,25], and magnetic nanoparticles as a contrast agent for MRI exams in three studies (23%) [32,33,34]. Regardless of the tracers used for labeling, this process allows HSC tracking and engraftment evaluation.

#### 3.4.1. Labeling of HSCs with Radioisotopes/Radiopharmaceuticals

HSC labeling with radioisotopes was reported in six studies [26,27,28,29,30,31], and all of the details of this process are described in Table 3. The common radioisotope used in three of the selected studies (50%) [29,30,31] was the 99mTc-exametazime (99mTc), which has a half-life of 6.03 h and it is associated with hexamethylpropyleneamine oxime (HMPAO) molecules (Ceretec, GE Healthcare). Regarding the labeling process, two of these studies [29,30] incubated 2 × 10^6^ cells with 37 MBq 99mTc-HMPAO for 30 min and the yield was 15 ± 3% after labeling, with an activity to administration of 5.55 MBq; the study by Lange [31] has already labeled with the range of 550–583 MBq 99mTc-HMPAO, and after labeling, the radioisotope activity was between 669–1350 MBq.

Another two studies [26,28] used the zirconium-89 (89Zr) radioisotope obtained by Cyclotron, and one study [28] added an oxine molecule to the radioisotope, which has a half-life of 78.4 h, but the labeling process differed between studies. The study by Asiedu [28] incubated 10^6^ cells with 0.01–5.55 MBq 89Zr-oxine for 20 min, resulting in a cell-associated radioactivity of 0.0036–1.7 MBq and a yield of 26–30%; the study by Pantin [26] already incubated 2 × 10^8^ cells with 0.37 MBq 89Zr for 30 min, and after labeling, the cell activity was 0.28 MBq. The study by Faivre [27] used [18F] fluorodeoxyglucose (18F-FDG) obtained by Cyclotron, which has a half-life of 1.83 h, and labeling was performed with a radioisotope activity ranging from 301.8 to 945.9 MBq after labeling of 5–10 MBq, with a purity of 94.6 ± 6%.

#### 3.4.2. Labeling of HSCs with Fluorophore

HSC labeling with fluorophore was reported in four of the selected studies [21,23,24,25], although in the study by Lopes [22], genetically modified animals were used that express green fluorescent protein (GFP), which, thus, required no labeling process, as described in Table 4. Most of the selected studies used fluorophore that emits radiation in the range of visible light wavelength, with the exception of the study by Ushiki [25], which used fluorophore that emits in the near-infrared fluorescence wavelength. The agent concentration and time of incubation during the labeling process varied among the studies, and two of the studies [22,25] reported a cellular toxicity evaluation after labeling, using mainly the flow cytometry technique.

#### 3.4.3. Labeling of HSCs with Nanoparticles

HSC labeling with magnetic nanoparticle agents was reported in three of the studies (14%) [32,33,34] (Table 5). Of these studies, two [32,33] used commercial nanoparticles such as Feridex^®^ [32] manufactured by Berlex Laboratories (Montville, New Jersey, USA), Resovist^®^ [33] manufactured by Bayer Schering Pharma AG (Berlin, Germany), and Endorem^®^ [33] manufactured by Guerbet S.A. (Roissy, France), with the exception the study by Sweeney [34], which synthetized nanoparticles of Gadolinium oxide. The nanoparticle characteristics of the selected studies were similar; the nanoparticles’ core sizes ranged from 3 to 5 nm [32,33], the coating was reported mainly with dextran [32,33], the hydrodynamic diameter ranged from 60 [33] to 180 [33,34], being monodisperse (polydispersity index-PDI lower than 0.5) in most of the studies [32,33]. The labeling process reported in the selected studies was performed with mainly two types of transfection agents (protamine sulfate [32,34] and lipofectamine [33]), and the nanoparticle concentration varied between 25 [33] and 125 [34] µg/mL, with mainly 4 h of incubation [33,34], with the exception of the study by Bengtsson [32], which used overnight incubation. Most of the selected studies [32,33] used labeling evaluation by Prussian blue. 

### 3.5. Administration of HSCs Labeled/Transfected in the Bone Marrow Transplant Model

Table 6 describes that most of the selected studies (57%) [9,11,16,18,19,20,25,27,28,32,33,34] used mice as the animal of choice as the cell receptor in the bone marrow transplant model. Among the different mice species (Nod scid gamma mouse [9,11,16,27], C57Black 6 [19,20,28,32], Balb/c [18,33], and 129/SvJ [34]), the majority were female [18,27,28,32], mainly 8-weeks-old (ranging from 6 to 12 weeks). Following the popularity of the use of mice, four further studies (19%) [17,21,23,24] used zebrafish in early development [23] (3-5 h post-fetal) and adult (48 h [24] and 52 h [21]) stages, and two more studies (10%) [29,30] used male Lewis rats (7-weeks-old). Of the other studies, one used White Axolt Salamanders [22] that were more than 12 weeks of age, one used male Beagle Dogs [31] that were 1–3-years-old, and one used domestic swine [26] of both genders weighing between 38 and 70 kg. Some animals of the selected studies were irradiated by Cesium-137 in six studies [11,16,18,25,27,34] and by X-rays in four studies [16,17,29,30], exposing them to total body irradiation (TBI) in most of the studies (71%) [9,11,16,17,18,19,20,22,25,27,28,29,30,31,32,34] with varied doses of irradiation. The most used dose was 9.5 Gy [19,22,28,29,30,32], ranging from 2.25 [27] to 20 [17] Gy, although for the immunosuppressed animals (NSG mice), the dose used was lower than 3 Gy [9,11,16,27]. The study by Lin used radiation for spinal cord ablation and also used the associated O6-benzylguanine (BG) and 1,3-bis(2-chloroethy1 (BCNU) for non-myeloablative conditioning. A few studies reported a dose rate of irradiation that ranges between 0.9 [29,30] and 2.7 [17] Gy/min.

After irradiation or not, the animals received the cell transplantation. The allogeneic type was most used (57%) [16,17,18,19,20,22,25,28,29,30,32,34] between the selected studies, followed by the xenogeneic type (38%) [9,11,21,23,24,26,27,33], and the study by Lange [31] used autologous transplantation. The time of cell implantation varied among studies; five studies [27,28,29,30,34] performed the implantation after 24 h of animal irradiation, three studies [9,18,25] used the acute stage between 1 and 8 h, and two studies [17,18] performed the implantation after 48 h of animal irradiation. The dose of cells implanted varied among studies; a dose 10^6^ cells was more common in the selected studies [16,18,27,29,30,31,32,33,34], ranging from 5 × 10^2^ (in zebrafish embryos) [22] to 10^8^ cells (in porcine) [26]. These cells were administered systemically in most of the selected studies (81%) [9,11,16,17,19,20,22,24,25,26,27,28,29,30,32,33,34] (81%), with the exception of the study by Lange [31], which used intrabone administration. Another four studies [16,25,26,29] compared systemic to intrabone administration, and in the studies that used zebrafish as a model, the cells were administered by yolk sac of Duct of Cuvier [21,24], intracardially [17], or retro-orbitally [21,34]. In vitro graft analysis was performed in 18 studies (86%) [9,11,16,17,18,19,20,22,23,24,25,27,28,29,30,32,33,34], and the technique most used for this analysis was the flow cytometry technique (FCT), as reported in 14 of the studies (67%) [9,11,16,17,18,19,22,23,24,25,27,28,33,34]. The efficiency of the graft was evaluated in 13 of the selected studies (61%) [9,11,16,17,19,20,23,25,27,28,30,33,34], and the results reported that before one month of implantation, cell renewal was very low (less than 20%) [23,27,30,33,34], but after three months, more than 50% efficiency of the graft was reported [11,16,19,20,28], with the exception of the study by Wang [9] that reported 1.3% graft efficiency at 3.5 months. In the studies that used allogeneic cell transplant [16,19,20,25,28], the efficiency of the graft was greater than xenogeneic cell transplant [9,11,23,27,33], achieving high efficiency in a short time.

### 3.6. Imaging Techniques Used in HSC Migration Homing and Tracking Analysis

HSC tracking was evaluated by different noninvasive imaging techniques after their transplant in the animals. The technique most used in the selected studies was bioluminescence in seven of the studies (33%) [9,11,16,17,18,19,20], followed by PET/SPECT used in six studies (29%) [26,27,28,29,30,31], fluorescence imaging in five studies (24%) [21,22,23,24,25], and MRI in the other three studies (14%) [32,33,34].

#### 3.6.1. Bioluminescence

Table 7 shows that the bioluminescence images were acquired after luciferin administration; this substrate was applied in most of the studies (86%) by intraperitoneal administration [11,16,17,18,19,20], using a dose that ranged from 0.75 to 150 mg/kg with different time intervals (2-15 min) to begin the image acquisition. The Xenogen IVIS system equipment was used in all studies, and although the model varied, the same software was used to process the images; that is, the Living Image software. The selected studies that used mice [9,11,16,18,20] acquired images mainly in the dorsal and ventral animal positions; the study by Steiner [11] also used the lateral position. Among the few BLI acquisition parameters reported by the selected studies, the exposure time used in the image acquisition varied among studies in accordance with the signal finding, but three studies [9,11,16] reported a decrease in the exposure time over the weeks (3 to 1 min/1 s) due to rise of signal intensity. Binning acquisition parameters was reported only in the study by Astuti [17], with the 4 or 8 values. The HSC tracking was performed from 1 day after bone marrow transplant in three studies (43%) [16,17,20], and in another four studies [9,11,18,19], bioluminescence imaging only was detectable at the end of the first week. Bioluminescence imaging was used for follow-up evaluation of longitudinal HSC tracking in all studies [9,11,16,17,18,19,20].

#### 3.6.2. PET/SPECT

Of the six studies that tracked cells with radioisotope agents (Table 8), three of them (50%) [29,30,31] used the single-photon emission computed tomography (SPECT) technique with 99mTc-HMPAO by gamma camera equipment, two [29,30] used a window energy of 5% centered over 140 KeV and dynamic acquisition, and the final study [31] used whole-body acquisition. The time of the cell homing evaluation of these studies was 30 min [29,30] and 24 h [31], detecting uptake distribution commonly to the heart, liver, lung, spleen, and other regions.

Another three studies (50%) [26,27,28] used hybrid positron emission tomography–computed tomography (PET-CT) scanning by a micro-PET-CT scanner [27,28] or a Philips clinical PET-CT scanner [26]. The studies by Asiedu [28] and Pantin [26] used 89Zr radioisotopes, a whole-body image with a 400 to 700 KeV energy window, and the homing evaluation was reported in intervals of 2-4 h or 1–7 days in the former study and 5-15 h in the latter study. The study by Faivre [27] used 18F-FDG with dynamic image acquisition and a 511 KeV energy window for 3 h of homing evaluation. Regarding the uptake distribution of cells after transplantation, the selected studies [26,27,28,29,30] that implanted cells by systemic administration (intravenous) reported greater concentrations of HSCs labeled in the lung. In two studies [26,29] that implanted the cells intraosseously, the cell distribution to the lung was lower, and posterior cell migration for other organs was reported.

#### 3.6.3. Fluorescence Imaging

Fluorescence imaging was performed in five of the studies [21,22,23,24,25] for HSC tracking analysis after the bone marrow transplant (Table 9). In the studies that used zebrafish or axolotl models, microscopy was used for imaging acquisition [21,22,23,24], and in the study by Ushiki [25], which used mice as receptors, the HSCs were tracked using the IVIS Spectrum system (Xenogen, Alameda, CA, USA). FLI acquisition parameters were reported only in the study by Ushiki [25]. Two studies reported the acute evaluation of HSCs after transplantation (until 24 h), but the study by Lopes reported longer homing evaluation (until 6 days) with positive detection until three days. Only the study by Ushiki [25], which used mice as cell receptors, was the uptake distribution reported in the bone marrow, lung, spleen, liver, and kidney, corroborating the ex vivo analysis [25]; in the studies that used zebrafish, the uptake distribution was reported mainly for the tail [21,23,24].

#### 3.6.4. Magnetic Resonance Imaging

Of the selected studies, MRI was performed in three studies [32,33,34] to evaluate HSC tracking after the bone marrow transplant (Table 10). The study by Niemeyer [33] only evaluated the acute stage from 2–24 h, using low magnetic field equipment (1.5 Teslas), but the other two studies [32,34] reported a long evaluation, from 3 h [34] to 14 days [32], using equipment with a higher magnetic field (4.7 [34] and 17.6 [32] Teslas). These three studies [32,33,34] reported uptake distribution for the bone marrow, liver, and spleen. MRI acquisition parameters used in the selected studies showed that the T2 weighted image was widely used with variations in the field of view, matrix, slice thickness, and others. In addition, this technique was the most detailed in terms of image acquisition parameters among the selected studies.

All aspects of the analysis of the noninvasive tracking of hematopoietic stem cells in a bone marrow transplant model are shown in Figure 2.

## 4. Discussion

The results of this review show an increase in using noninvasive imaging techniques in HSC tracking in the bone marrow transplant model. The most selected studies used bioluminescence, followed by fluorescence, PET/SPECT, and MRI. The use of noninvasive imaging techniques can reveal that hematopoietic activities in both steady-state and pathological conditions are dynamic, and that their sequence is regulated spatiotemporally by interaction with the niche [55]. Successful clinical trials show that the engraftment depends on adequate HSC numbers, their homing, and the subsequent short-term and long-term engraftment of these cells in the niche (bone marrow). Enhancing the homing capability of HSCs could have a great impact on improving transplantation procedures and patient survival [56]. The main challenge of leukemia or the dysfunctional or depleted bone marrow preclinical model is the mimicking of the bone marrow niche or the environment. Experimental findings in mice often correlate with human biology and, as such, they serve as a research stand-in for human patients [57]. In this review, most of the selected studies used mice as the animal model, and regarding the type of graft, allogeneic cells were the most used among the studies. In exploring this aspect, it was reported that when comparing allogeneic and xenogeneic grafts, using the cell route by the tail vein, the allogeneic cells arrive at the niche faster (3 h after transplantation) than in the xenogeneic cells (15 h after transplantation), and in a greater amount, being detectable in the endosteal region of the femur (13% of cell quantity) and in the central marrow region (58% of cell quantity), associated with efficiency and marrow repopulating ability [58].

Irradiation is an important aspect of HSC homing analysis. The study by Xie [59] showed that only 7% of cells after 6 h of transplantation arrive at the bone marrow niche compared to the irradiated hosts, which obtained better efficiency with 35% of cells detectable in the niche. In addition, immunoassays identified the endosteal zone of irradiated recipients as a site of increased HSC proliferation after transplantation. In this review, it was observed that most rodent models of the selected studies applied bone marrow irradiation with cesium-137 or X-ray, and cell homing analysis was possible and detectable by noninvasive image and other techniques, particularly the flow cytometry technique.

Optical imaging certainly holds utility in assessing hematopoietic stem cell tracking of xenografts, allografts, autologous grafts applied in the bone marrow transplant model [35]. The same study reported that bioluminescence imaging is certainly more sensitive than fluorescence imaging. However, one strength of fluorescent fluorophores is the ability to perform high-resolution microscopy of GFP-expressing cells. As a result, if tracking of small numbers of cells at high magnification and perhaps in real-time is desired, fluorescence-based methods can be considered. Besides these techniques, this review reported the used of other techniques, such as PET/SPECT and MRI in cell homing, where PET/SPECT has a small temporal window in signal detection and MRI requires a magnetic contrast agent that may interfere with cellular growth biology and the interaction of the environment in a niche (bone marrow), decreasing the sensitivity of this technique.

Most of the selected studies of this review used bioluminescence due to its reported high sensitivity, with a wide temporal detection window (0 days to 1 year), but a limitation of this technique is that the light propagation through tissue currently restricts the application of bioluminescence imaging to small animals, where the signal can easily penetrate at all depths, and it is a technique that has not yet been well studied in vivo [9]. Another prevalent technique used by six of the selected studies was PET/SPECT imaging, which provides three-dimensional information with high sensitivity, excellent penetration depth, and a capability for whole-body imaging, but which requires radioisotopes. An advantage of this technique is that it can also be applied in small animals as well as in humans, and the disadvantages include the high cost of the cyclotron and limited temporal follow-up due to the radioisotopes’ half-life—a factor that was reported in the selected studies, where cell homing was commonly evaluated at 15 h. In addition, in the selected studies, the radioisotope most used was the 99mTc-HMPAO, which has high translation capacity, but it is important to bear in mind the potential effects of radiation on therapeutic/biology cell function [60].

The fluorescence imaging technique has a high sensitivity (lower than bioluminescence), a high temporal resolution, a low cost, and it is activatable, but the disadvantages in relation to the other image techniques described before include the attenuation of sensitivity by overlying tissues, low spatial resolution, and poor penetration depth. Therefore, this technique is the most important imaging modalities for live-cell imaging at single-cell resolution using light sources to excite the fluorescent molecules [61]. In this review, the fluorescence technique observed HSC homing into the niche up until 24 h.

MRI has a high spatial and temporal resolution, no tissue penetrating limit, and no radiation, but the disadvantages include relatively low sensitivity and low contrast, requiring a high load of cells with a magnetic label and comparatively long imaging times, with the possibility of affecting cell viability or biological interaction with the contrast agent [43,62,63,64]. In this review, three studies used HSC tracking analysis by MRI with different magnetic fields, in which the highest magnetic field (17.6 Teslas) provided longer follow-ups (14 days after transplantation) when using low nanoparticle concentrations and incubation times in the cell labeling process. Our group provided evidence, in previous studies, that a low nanoparticle concentration in the labeling process was enough to detect stem cell migration homing and tracking. Additionally, nanoparticle sizes of between 35 and 200 nm used in our previous studies showed magnetic characteristics adequate for detection by MRI [65,66,67]. This review corroborates the nanoparticle size findings among the selected studies (60 to 180 nm).

Regarding the molecular imaging techniques approaches in this review, their physical principles, applicability, advantages, and limitations (Figure 3) [43,46,47,68,69,70,71,72,73], show a wide potentiality not only for in vitro studies and pre-clinical applications but also in the translation of some techniques in clinical studies, such as nuclear images (PET and SPECT) and MRI [47,73]. Comparing the molecular imaging technique characteristics and the HSC tracking evaluation shown in Figure 3, the optical techniques allowed a live cell tracking for a long period of time by BLI (at 1 day to 12 months)) [9,11,16,17,18,19,20] and short time by FLI (at 30 min to 6 days) [21,23,24,25], but in the latter, the signal is not necessarily from living cells. In nuclear techniques, the time window varies with the radioisotopes used due to their half-life, in SPECT the radioisotopes have a shorter half-life, and therefore, the time allowed for tracking the cells was also shorter (at 30 min to 1 day) [29,30,31], compared with cell tracking made by PET (at 1 h to 7 days) [26,27,28], in which the radioisotopes used have an average half-life of 78.4 h, as is the case with ^89^Zr. MRI has high spatial resolution and a wide temporal window for cell tracking; however, for all this, a high nanoparticle load is required for its detection, as its sensitivity is low (10^−3^–10^−5^ mol/L) [72] compared to the other techniques, in addition to decreasing the signal temporally due to cell division, making it difficult to keep track of the cells [74].

Therefore, for preclinical studies of the cell transplantation model, all molecular imaging techniques covered in this review have good applicability in cell tracking for early assessment, but the BLI optical technique stands out in the prolonged tracking assessment [11]. In the clinical application of oncological diseases, the PET nuclear technique and MRI would be the most suitable for early assessment of cell tracking [26,27,28,33,34]. However, cell graft was not evidenced by any of the molecular techniques mentioned in this review, and for cell graft to occur, we must have good control of cell tracking. The studies selected in this review used other techniques to quantify chimerism upon transplantation. The technique most used to access chimerism in the studies selected in this review was the FCT to determine the percentage of grafted donor cells grafted using specific cell line markers such as CD45 expressed by all lymphocytes [9,11,19,23,24,27,28,33], or GFP expressed by cells after genetic modification [17,22,25,34]. The CD45, in addition to making it possible to analyze the occurrence of engraftment, also allows checking the percentage of chimerism through discrimination of its two different alleles (CD45.1 and CD45.2), which are functionally identical. CD45.1+ donor cells can be readily detected when transplanted into CD45.2 mice, as reported in two studies in this review [19,28]. In these studies, the efficiency assessment was performed at early (2 to 4 days after implantation) [23,25,33] and late time points after transplantation (after 20 days to 365 days after implantation) [9,11,16,19,20,27,28]. The evaluation of cell grafting is one of the most relevant aspects of clinical application in cancer patients, as the biodistribution and efficiency of implantation are the targets for the success of the treatment, aspects that still have many gaps in clinical research.

Successful transplantation depends on the formation of engraftment, in which donor cells are integrated into the recipient’s cell population, as well as a supportive hematopoietic stromal microenvironment [75]. Evaluation of this was performed in most of the selected studies, which showed that the studies that used allogeneic cell transplantation had greater graft efficiency in a shorter time period than xenogeneic transplantation in the experimental model. Another relevant aspect of the success of transplantation is the functionality of cells after the graft, because graft failure and poor graft function are important issues in the current comprehensions of the interaction between the immune and hematopoietic compartments in these conditions [76]. Therefore, besides the quantitative analysis of the graft, the graft functionality should be adequate in the evaluation of graft efficiency, but this aspect was poorly approached in most of the selected studies.

Unfortunately, there are not enough clinical HSCs available to set up phase I/II clinical trials to test the tracking of these new cells by noninvasive imaging techniques. Most investigators conducting such trials are “wed” to their personal favorite procedure. If, in the future, we can deal with this problem and find means for additional clinical efforts, it is possible that several new procedures could be used together [34], such as hybrid equipment or imaging systems. This, however, adds additional logistical problems versus the use of one procedure alone, such as requiring multifunctional probes, smart processes, and improvement of the technical limitations of imaging equipment [77]. Therefore, there is viability to translate these experimental findings into bed-side application. The cell tracking improvement of niches provides a reduction in therapeutic time with high efficiency of hematopoietic cellular renewal.

## Figures and Tables

**Figure 1 cells-09-00939-f001:**
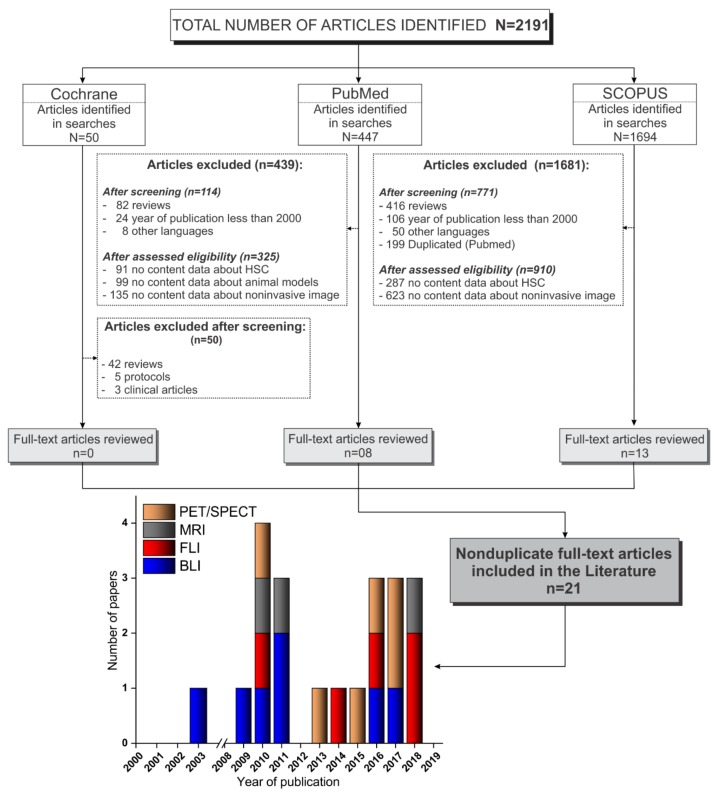
Flowchart corresponding to the stages of the PRISMA guidelines [54] of the article screening process for inclusion in this review. Abbreviations: HSC, hematopoietic stem cell; SPECT, single-photon emission computed tomography; PET, positron emission tomography; MRI, magnetic resonance imaging; FLI, fluorescence; BLI, bioluminescence.

**Figure 2 cells-09-00939-f002:**
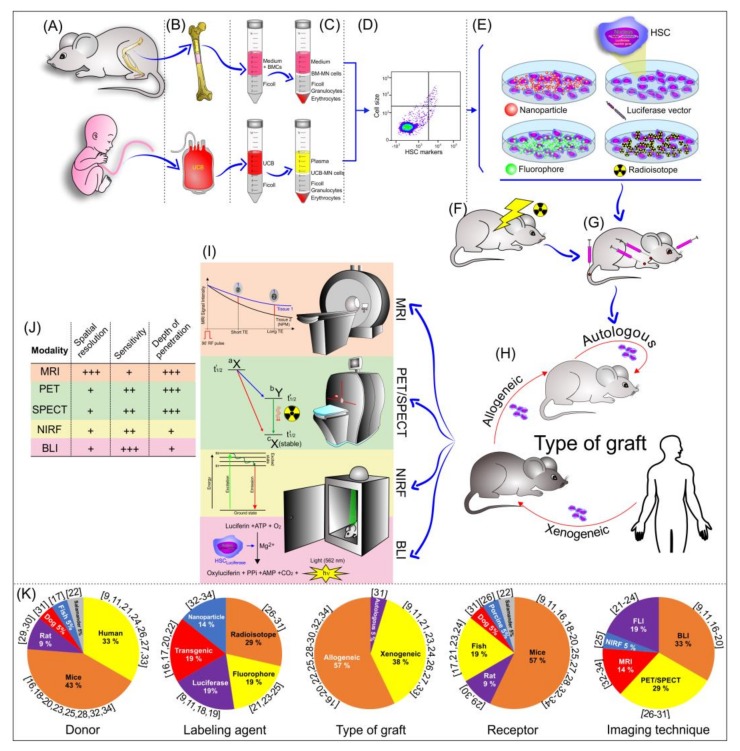
Schematic illustration of the bone marrow transplantation process, from the isolation of HSCs to their implantation, and tracking evaluation by noninvasive imaging techniques. (**A**) Main cell donors; (**B**,**C**) main cell sources for HSC extraction; (**D**) HSC characterization after isolation; (**E**) contrast agent used in the HSC labeling; (**F**) animal pre-condition (irradiation) before HSC transplantation; (**G**) cell route used in the transplantation; (**H**) type of graft; (**I**) noninvasive imaging technique used in HSC tracking; (**J**) imaging modality features; and (**K**) the graphic of the percentual distribution of the main element analyzed in the systematic review.

**Figure 3 cells-09-00939-f003:**
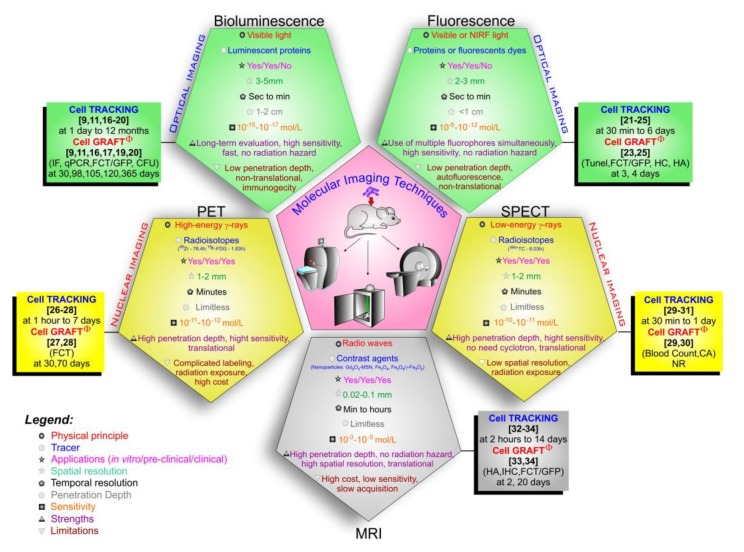
Molecular imaging modalities comparing their techniques features, tracers, applications, highlighting the follow time of each technique in the HSC tracking and the analysis of cell graft by other techniques after transplantation. Abbreviations: sec—seconds, min—minutes, cm—centimeters, mm—millimeters, mol—mole, L—liter, NIRF: near-infrared fluorescence; GFP: green fluorescent protein; qPCR: quantitative polymerase chain reaction; TUNEL: terminal deoxynucleotidyl transferase dUTP nick end labeling; CFU: colony forming units; FCT: flow cytometry technique; IHC: immunohistochemistry; IF: immunofluorescence; HC: histochemical; HA: histological analysis; CA: compartmental analysis; MSN: mesoporous silica nanoparticle; Gd_2_O_3_: Gadolinium oxide; Fe_3_O_4_: magnetite, γ-Fe_2_O_3_: maghemite. Note: φ the cell graft was analyzed by other techniques, without being the molecular imaging mentioned in the Figure.

**Table 1 cells-09-00939-t001:** Extraction and isolation of hematopoietic stem cells.

Ref.	Year	Extraction of HSCs						Isolation of HSCs			
		Donor	Source of Cells	Donor Age	Donor Gender	Harvest	Medium and Supplementation	Separation	Immunophenotypic Characterization	Technique for Sorting	Purity (%)
Parada-Kusz et al. [23]	2018	Mice (C57BL/6 or CByJ.B6- UBI-GFP)	BM	6-8 w	M-F	Maceration	PBS (0.5% BSA/BFS, 2 mM EDTA)	NR	CD45+; c-kit+; CD11b+	MACS	NR
Sweeney et al. [34]	2018	Mice (129/SvJ)	ESC	NR	NR	Differentiation	StemPro34 with cytokines	NR	CD41+; CD45+; Sca-1+ c-Kit+	NR	NR
Asiedu et al. [28]	2017	Mice (C57BL/6 CD45.2)	BM	NR	M-F	NR	NR	NR	NR	NR	NR
Lange et al. [31]	2017	Dog (Beagle)	BM	1–3 y	M	Aspirate	NR	Ficoll	CD34+	NR	NR
Astuti et al. [17]	2017	Fish (zebrafish ubi:luc)	WKM	NR	NR	Maceration	PBS	NR	NR	NR	NR
Saia et al. [20]	2016	Mice (Ubi-Luc2KI)	BM	8–10 w	NR	NR	NR	NR	Lin-	MACS	NR
Lopez et al. [22]	2014	Axolotl (GFP+; nucCherryRed+)	SP; LV	NR	NR	Maceration	APBS (5% FBS)	Ficoll	NR	FACS	NR
Sambuceti et al. [30]	2013	Rat (Lewis)	BM	7 w	M	Flushing	DMEN (10% FBS, UFH)	Ficoll	CD90+	FACS	NR
Lin et al. [18]	2011	Mice (Balb/c)	BM	6–8 w	F	Flushing	α-MEM (10% UFH)	NR	NR	NR	NR
Andrade et al. [16]	2011	Mice (FVB H-2Kq)	BM	NR	NR	NR	NR	NR	NR	NR	NR
Bengtsson et al. [32]	2011	Mice (TgCAGDsRed)	BM	NR	M	Flushing	NR	NR	lin- c-kit+ Sca-1+	MACS; FACS	NR
Ohmori et al. [19]	2010	Mice (C57BL/6 Ly5.1)	BM	NR	NR	NR	NR	NR	lin- c-kit+ Sca-1+	MACS; FACS	NR
Ushiki et al. [25]	2010	Mice (Balb/c nu/nu)	BM; SP	7–9 w	NR	NR	NR	Lympholyte	NR	MACS	NR
Massolo et al. [29]	2010	Rat (Lewis)	BM	7 w	M	Flushing	DMEN (10% FBS, UFH)	Ficoll	CD90+	FACS	NR
Hamilton et al. [21]	2018	Human	PB	18-40 y	M-F	NR	NR	Percoll	CD34+	MACS; FACS	NR
Faivre et al. [27]	2016	Human	CB	NR	NR	NR	NR	NR	CD34+	MACS	94.6
Staal et al. [24]	2016	Human	CB	NR	NR	NR	NR	Ficoll	CD34+	MACS; FACS	95
Pantin et al. [26]	2015	Human	PB	NR	NR	Apheresis	NR	NR	CD133+	MACS	NR
Niemeyer et al. [33]	2010	Human	CB	NR	NR	NA	Citrate buffer	Ficoll	CD34+	MACS	>95
Steiner et al. [11]	2008	Human	CB	NR	NR	NR	NR	Ficoll	CD34+	MACS; FACS	>95
Wang et al. [9]	2003	Human	CB	NR	NR	NR	NR	NR	CD34+ CD38-; CD34+	MACS; FACS	80-90

Abbreviations—Ref.: reference; BM: bone marrow; ESC: embryonic stem cell; WKM: whole kidney marrow; SP: spleen; LV: liver; PB: peripheral blood; CB: cord blood; w: weeks; y: years; M: male; F: female; PBS: phosphate-buffered saline; APBS: 0.753x phosphate-buffered saline; DMEN: Dulbecco’s modified Eagle’s medium; α-MEM: minimum essential medium Eagle—alpha modification; FBS: fetal bovine serum; BSA: bovine serum albumin; EDTA: ethylenediaminetetraacetic acid; UFH: unfractionated heparin; Lin: lineage; FACS: fluorescence-activated cell sorting; MACS: magnetic-activated cell sorting; NR: not reported.

**Table 2 cells-09-00939-t002:** Luciferase transduction in hematopoietic stem cells.

Ref.	Tracking Agent	Vector	Cell Dose	Transfection Agent	MOI	Medium Supplemented	Cytokines SDT (ng/mL)	Incubation (Hours)	Evaluation
Lin et al. [18]	FLUC	Lentiviral plasmid, pCSO-rre-cppt-MCU3-LUC	NR	Polybrene 8 μg/mL	0.5–1	DMEN; Iscove’s MEM (20% FBS)	mIL-3: 20; mIL-6: 50; rSCF: 50	48	FCT; CFU; PCR
Ohmori et al. [19]	LUC	LentiLox vectors	NR	Polybrene 8 μg/mL	20	StemPro 34 SFM	SFC: 100; TPO: 100; IL-6: 100; Flt-3L: 100; sIL-6R: 200	24	FCT
Steiner et al. [11]	FLUC	Plasmid pHIV-GFPFFLuc	3-8 × 10^5^ per well	Retronectin 100 μg/mL	NR	α-MEM (10% FCS; 2mM L-Gln)	NR	Overnight	FCT
Wang et al. [9]	FLUC	Lentiviral vector SMPU-R-MNCU3-LUC	1–10 × 10^4^ per plate; 3–30 × 10^2^ per well	NR	NR	NR	IL-3: 5; IL-6: 16.5; SCF: 25	24 (2 cycles)	IHC

Abbreviations—Ref.: reference; FLUC: firefly luciferase; LUC: luciferase; MOI: multiplicity of infection; DMEN: Dulbecco’s modified Eagle’s medium; IMEN: Iscove’s modified Eagle’s medium; α-MEM: minimum essential medium Eagle—alpha modification; FBS: fetal bovine serum; FCS: fetal calf serum; SFM: serum-free media; Gln: Glutamine; Cytokines SDT: cytokines stimulation during transduction; mIL: murine interleukin; IL: interleukin; SCF: stem cell factor; TPO: thrombopoietin; Flt-3L: fms-like tyrosine kinase 3 ligand; sIL-6R: soluble IL-6 receptor; FCT: flow cytometry technique; CFU: colony forming unity; PCR: polymerase chain reaction; IHC: Immunohistochemistry; NR: not reported.

**Table 3 cells-09-00939-t003:** Labeling of hematopoietic stem cells with radioisotopes/radiopharmaceuticals.

Ref.	Radioisotope	Radiopharmaceuticals	Half-Life (h)	Manufacture	Labeling Activity (MBq)	Activity after Labeling (MBq)	Cell Dose	Incubation (min)	Yield (%)
Asiedu et al. [28]	89Zr	Oxime	78.4	Cyclotron	0.01–5.55	0.0036- 1.7	1 × 10^6^	20	26-30
Lange et al. [31]	99mTc	HMPAO	6.03	Ceretec, GE Healthcare	550–583; 669–1350	NR	NR	NR	NR
Faivre et al. [27]	18F	FDG	1.83	Cyclotron	301.8–945.9	5–10	NR	30	94.6 ± 6
Pantin et al. [26]	89Zr	NA	78.4	Cyclotron	0.37	0.28	2x10^8^	30	NR
Sambuceti et al. [30]	99mTc	HMPAO	6.03	Ceretec, GE Healthcare	37	5.55	2x10^6^	30	15 ± 3
Massolo et al. [29]	99mTc	HMPAO	6.03	Ceretec, GE Healthcare	37	5.55	2x10^6^	30	15 ± 3

Abbreviations—Ref.: reference; 89Zr: radioactivity zirconium isotope; 99mTc: radioactivity technetium metastable isotope; 18F-FDG: 18Fluoride - fluorodeoxyglucose; HMPAO: hexamethylpropyleneamine oxime; GE: General Electric; min: minute; h: hour; MBq: Mega Becquerel; NR: not reported; NA: not applicable.

**Table 4 cells-09-00939-t004:** Labeling of hematopoietic stem cells with fluorophore.

Ref.	Agent	Manufacture	Excitation/ Emission (nm)	Concentration	Incubation (min)	Evaluation
Parada-Kusz et al. [23]	CellTrace™ Violet Cell Proliferation Kit	Thermo Fisher Scientific Inc.	405/450	5 µM	20–30	NR
	CellTrace™ CSFE	Thermo Fisher Scientific Inc.	495/519	5 µM	20–30	NR
Hamilton et al. [21]	Fluorocein	NR	495/519	NR	10	NR
Staal et al. [24]	PKH26 (#PKH26GL)	Sigma Aldrich	551/567	2 µM	2	NR
Lopez et al. [22]	GFP*	NA	475/509	NA	NA	FCT; PCR
Ushiki et al. [25]	Cy5.5	GE Healthcare UK Ltd., Buckinghamshire, UK	675/694	0.4 mg/mL	15	FCT
	AF750	Invitrogen, Eugene, OR, USA	749/775	0.1 mg/mL	15	FCT

Abbreviations—Ref.: reference; CFSE: 5-(6)-carboxyfluorescein diacetate succinimidyl Ester; GFP: green fluorescent protein; Cy5.5: cyanine 5.5; AF750: Alexa Fluor 750 carboxylic acid, succinimidyl ester; PCR: Polymerase chain reaction; nm: nanometer; µM: micromolar; mg: milligrams; mL: milliliters; min: min; FCT: flow cytometry technique; NR: not reported; Note—* animals genetically modified that express GFP.

**Table 5 cells-09-00939-t005:** Labeling of hematopoietic stem cells with nanoparticles.

Ref.	Particle	Size Core (nm)	Coating	D_H_ (nm)	Manufacture	PDI	Transfection Agent	Concentration (µg/mL)	Incubation (h)	Evaluation
Sweeney et al. [34]	Gd_2_O_3_-MSN with pores 24Å	NR	Pore functionalized with: (a) TRITC; (b) FITC; (c) TRITC and PEG	177	Synthetized	0.535	Polybrene or protamine sulfate	125	2–4	Fluorescent microscopy
Bengtsson et al. [32]	Magnetite	4.8	Dextran	80–150	Feridex (Berlex Laboratories, Montville, NJ, USA)	0.29	Protamine sulfate	NR	Overnight	Prussian blue
Niemeyer et al. [33]	Magnetite/maghemite	3-5	Carboxydextran	60	Resovist (Bayer Schering Pharma AG, Berlin, Germany)	0.207	Lipofectamine	25	4	Prussian blue
	Magnetite	4.8	Dextran	120–180	Endorem (Guerbet S.A., Roissy, France)	0.266	Lipofectamine	25	4	Prussian blue

**Abbreviations—**Ref.: reference; D_H_: Hydrodynamic diameter; MSN: Mesoporous silica nanoparticle; Gd_2_O_3_: Gadolinium oxide; TRITC: tetramethyl rhodamine isothiocyanate; FITC: fluorescein isothiocyanate; PEG: Poly(ethyleneglycol); PDI: polydispersity index; NR: no reported; nm: nanometer; Å: ångström; ∝g: micrograms; mL: milliliters; h: hours.

**Table 6 cells-09-00939-t006:** Administration of hematopoietic stem cells (HSCs) labeled/transfected in bone marrow transplant model.

Ref.	Animal Receptor			Irradiation				Cells Transplantation					Graft Assessment	Graft Efficiency (% or Number of Cells) At Time
	Specie	Age	Gender	Conditioning	Source	Dose (Gy)	Dose-Rate (Gy/min)	Type of Graft	Delay for Cells Infusion (Hours)	Cell Dose	Route	Vehicle		
Parada-Kusz et al. [23]	Zebrafish (nacre^−/-^)	3–5 h	NR	NA	NA	NA	NA	Xenogeneic	NA	1-6 × 10^4^	Blastoderm	PBS	FCT (CD45); TUNEL	0 at 4 d
Sweeney et al. [34]	Mice (129/SvJ)	NR	NR	TBI	137-Cs	2 × (7–8)	NR	Allogeneic	24	7.2 × 10^6^	IV (RO)	NR	FCT (GFP)	0 at 20 d
Hamilton et al. [21]	Zebrafish (Tg-kdrl:EGFP)	52 h	NR	NA	NA	NA	NA	Xenogeneic	NA	NR	DC yolk sac	NR	NA	NA
Asiedu et al. [28]	Mice (C57BL/6 CD45.1)	8–12 w	F-M	TBI	NR	9.5	NR	Allogeneic	24	2 × 10^7^	IV	NR	FCT (CD45.1, CD45.2, linage specify)	85 at 70 d
Lange et al. [31]	Dog (Beagle)	1–3 y	M	NA	NA	NA	NA	Autologous	NA	1.5 × 10^6^/kg	IB (humerus)	NR	NA	NR
Astuti et al. [17]	Zebrafish	NR	NR	TBI	X-rays	20	2.7	Allogeneic	48	5–50 × 10^4^	IV (IC)	NR	IF, FCT (GFP)	306.5 ± 136.6 cells at 14 d
Saia et al. [20]	Mice (B6 albino)	NR	NR	TBI	NR	6.5	NR	Allogeneic	NA	5 × 10^5^	IV (tail)	NR	qPCR (LUC2)	53 at 120 d
Faivre et al. [27]	Mice (NSG)	8 w	F	TBI	137-Cs	2.25	NR	Xenogeneic	24	2 × 10^6^	IV (tail)	Saline	FCT (CD34, CD45)	CD34: 1.5 ± 0.6; CD45: 20 ± 6 at 30 d
Staal et al. [24]	Zebrafish (casper Fli-GFP)	48 h	NR	NA	NA	NA	NA	Xenogeneic	NA	0.5–5 × 10^2^	DC yolk sac IV (RO)	NR	FCT (CD3, CD14, CD34, CD38, CD45)	NR
Pantin et al. [26]	Porcine (domestic swine)	NR	F-M	NA	NA	NA	NA	Xenogeneic	NA	2 × 10^8^	IV (jugular) IB (iliac crest)IA (iliac)	Saline	NA	NA
Lopez et al. [22]	Salamander (white mutant (d/d))	>12 w	NR	TBI	NR	9.5	NR	Allogeneic	NR	1-50 × 10^4^	IV (IC)	NR	FCT (GFP); TUNEL; HC	NR
Sambuceti et al. [30]	Rat (Lewis)	7 w	M	TBI	X-rays	9.5	0.9	Allogeneic	24	2 × 10^6^	IV (NA)	NR	CA	LV: 0.28 ± 0.18; SP: 0.19 ± 0.12; LG: 0.03 ± 0.42
Lin et al. [18]	Mice (Balb/c)	NR	F	BG + BCNU; TBI	137-Cs	7.5	NR	Allogeneic	48; 1–5	1–10 × 10^5^	NA	NR	FCT; CFU	NR
Andrade et al. [16]	Mice (NSG)	8–10 w	NR	TBI; RHLIR	137-Cs; X-rays	2.7	NR	Allogeneic	NR	1 × 10^4^; 1 × 10^5^; 1 × 10^6^	IV (tail) IB (femur)	NR	FCT (CD11b, Ly-6c/g, CD3ε, B220, NK1.1)	69 ± 5 at 98 d
Bengtsson et al. [32]	Mice (C57BL/6j)	6–8 w	F	TBI	NR	9.5	NR	Allogeneic	NR	1 × 10^3^; 1 × 10^6^	IV (RO)	NR	HA	NA
Ohmori et al. [19]	Mice (C57BL/6 Ly5.2)	8–12 w	NR	TBI	NR	9.5	NR	Allogeneic	NR	1 × 10^5^	IV (carotid)	NR	FCT; IF	58 at 30 d
Ushiki et al. [25]	Mice (Balb/c)	7–9 w	NR	TBI	137-Cs	8	NR	Allogeneic	6-8	1 × 10^7^	IV (tail) IB (tibia)	NR	FCT (eGFP); HA	43.5 at 3 d
Massolo et al. [29]	Rat (Lewis)	7 w	M	TBI	X-rays	9.5	0.9	Allogeneic	24	2 × 10^6^	IV (tail) IB (tibia)	Saline	Blood count	NR
Niemeyer et al. [33]	Mice (Balb/c)	NR	NR	NA	NA	NA	NA	Xenogeneic	NA	1-5 × 10^6^; 5 × 10^6^; 1 × 10^7^	IV (tail)	NR	HA; IHC; FCT (CD34, CD45, CD71)	0.08 at 1d
Steiner et al. [11]	Mice (NSG)	6–12 w	NR	TBI	137-Cs	2.7	NR	Xenogeneic	NR	8-35 × 10^4^	IV (tail)	NR	FCT (CD45, linage specify); IHC	39.6 at 365 d
Wang et al. [9]	Mice (NSG)	8–10 w	NR	TBI	NR	3	NR	Xenogeneic	2	4 × 10^4^; 1 × 10^5^	IV (tail)	NR	FCT (CD 45)	1.3 at 105 d

**Abbreviations**—Ref.: reference; h: hours; min: minute; w: week, y: year; mg: milligram; kg: kilogram; TBI: total body irradiation; RHLIR: right hind limb irradiation; BCNU: 1,3-bis(2-chloroethyl)-1-nitrosourea; BG: O6-benzylguanine; 137-Cs: Cesium-137; Gy: gray; IV: intravenous; IB: intrabone; RO: retro-orbital; IC: intracardiac; DC: duct of Cuvier; PBS: phosphate-buffered saline; qPCR: quantitative polymerase chain reaction; TUNEL: terminal deoxynucleotidyl transferase dUTP nick end labeling; CFU: colony forming units; FCT: flow cytometry technique; IHC: immunohistochemistry; IF: immunofluorescence; HC: histochemical; HA: histological analysis; CA: Compartmental analysis; LV: liver; SP: spleen; LG: lungs; NR: not reported; NA: not applicable; **Note****—**Mice (129/SvJ): mice have mutated CD23 and hyper IgE; Mice (NSG)—Nod scid gamma mouse; Mice—Balb/c: an albino mice; Mice (C57BL/6) wild type (expressing CD45.2) and congenic (expressing CD45.1).

**Table 7 cells-09-00939-t007:** HSCs migration homing and tracking by bioluminescence.

Ref.	Substrate	Dose (mg/kg)	Substrate Administration Route	Time Before Image (min)	Equipment	Software	Animal Position	Exposure Time (min)	Binning	Homing Evaluation Time	Outcome (Cells Migration)
Astuti et al. [17]	Luciferin	75#	IP	10–15	Xenogen IVIS50 system (Caliper Life Sciences)	Living Image	Dorsal and lateral	1	4; 8	98 d	**+**
Saia et al. [20]	Luciferin	80	IP	15	Xenogen IVIS Lumina system (Caliper, PerkinElmer)	Living Image	Ventral and dorsal	1	NR	1-10 d (20, 30, 40-120 d)	**+**
Lin et al. [18]	Luciferin	125	IP	7	Xenogen IVIS 200 Imaging system (Caliper Life Sciences)	Living Image	Ventral and dorsal	5	NR	146 d	**+**
Andrade et al. [16]	Luciferin	125	IP	8	Xenogen IVIS 100 imaging system (Caliber Life Sciences)	Living Image	Ventral and dorsal	3-1	NR	1-65 d	**+**
Ohmori et al. [19]	Luciferin	1.5^§^	IP	NR	IVIS Imaging system (Xenogen, Alameda, CA)	Living Image	Ventral	NR	NR	3, 7, 14, 21, …, 256 d	**+**
Steiner et al. [11]	Luciferin	150	IP	NR	Xenogen-IVIS Imaging system (Caliper Life Sciences Hopkinton, MA)	Living Image	Ventral, dorsal and lateral	3 -1 s	NR	7, 14 d (3, 6, 12 months)	**+**
Wang et al. [9]	Luciferin	125	IV	2	IVIS 3-D optical imaging system (Xenogen, Alameda, CA)	Living Image	Ventral and dorsal	3-1	NR	1, 8 d (7-15 weeks)	**+**

**Abbreviations**—Ref.: reference; IP: intraperitoneal; IV: intravenous; NR: not reported; mg: milligrams; kg: kilograms; μg: microgram; d: days; min: minutes. **Note**—Units: §: mg/body. #: μg/body.

**Table 8 cells-09-00939-t008:** HSCs migration homing and tracking by PET/SPECT.

Ref.	Image Modality	RP	Equipament	Reconstruction Parameters	Type of Image	Energy Window(KeV)	Image Acquisition Time	Homing Evaluation Time	Uptake Distribution	Outcome (Cells Migration)
Asiedu et al. [28]	PET-CT	89Zr-oxime	MicroPET-CT (BioPET, Bioscan)	3D; Iterative reconstruction	Wholebody	400–700	5 min per bed for 4 h; 6.5, 7.5, 12.5, 15.5 min per bed for 1, 2, 5,7 d	0, 2, 4, 24, 48 h (5, 7 d)	Bone marrow, spleen and liver	**+**
Lange et al. [31]	SPECT	99mTc-HMPAO	PRISM 2000 XP gamma camera (Phillips, Hamburg, Germany)	NR	Wholebody	140 ± 5%	NR	1, 6, 24 h	Bone marrow, lungs, ribs and spines	**+**
Faivre et al. [27]	PET-CT	18F-FDG	Micro-PET-CT (Inveon, Siemens)	OSEM 3D	Dynamic	511 ± 5%	Dynamic study for 2.5 h	0, 1, 2, 3 h	Lung, kidney, spleen, liver, femur and vertebrae	**+**
Pantin et al. [26]	PET-CT	89Zr	Gemini TF clinical PET/CT (Philips Healthcare, Andover, MA)	3D; Iterative reconstruction; Scatter and attenuation correction; Time of Flight; Spatial resolution: 4.8mm	Wholebody	400–700	NR	5, 10, 15 h	Lungs, bone marrow	**+**
Sambuceti et al. [30]	SPECT	99mTc-HMPAO	Gamma-camera (GE Millennium, Milwaukee, USA)	Parallel hole collimator; Image size 128 × 128 × 16	Dynamic	140 ± 5%	240 img of 0.5 sec, 60 img of 2 sec, 36 img of 10 sec, 5 img of 120 sec, 2 img of 300 sec	30 min	Heart, lung, liver, spleen	**+**
Massolo et al. [29]	SPECT	99mTc-HMPAO	Gamma-camera (GE Millennium, Milwaukee, USA)	Parallel hole collimator; Image size: 128 × 128 × 16	Dynamic	140 ± 5%	240 img of 0.5 sec, 60 img of 2 sec, 36 img of 10 sec, 5 img of 120 sec, 2 img of 300 sec	30 min	Heart, liver, spleen and forelimb, maxillary lymph node	**+**

**Abbreviations**—Ref.: reference; RP: Radiopharmaceuticals; PET-CT: positron emission tomography; SPECT: single photon emission tomography; 89Zr: radioactivity zirconium isotope; 99mTc: radioactivity technetium metastable isotope; HMPAO: hexamethylpropyleneamine oxime; 18F-FDG: 18Fluoride - fluorodeoxyglucose; 3D: three dimensional; OSEM: ordered subsets expectation-maximization; KeV: kiloelectron-volts; IBM-I: intra-bone hematopoietic stem cell transplantation condition I; IBM-II: intra-bone hematopoietic stem cell transplantation condition II; HSPC: hematopoietic progenitor cell; HSC: hematopoietic stem cell; IV: intravenous, IB: intrabone; img: image; mm: millimeter; sec: second; m: minute; d: day; NR: not reported.

**Table 9 cells-09-00939-t009:** HSC migration homing and tracking by fluorescence imaging.

Ref.	Tracking Agent	Equipment	Parameters	Homing Evaluation Time	Uptake Distribution	Outcome (Cells Migration)
Parada-Kusz et al. [23]	CellTrace™ Violet Cell; CellTrace™ CSFE	Epifluorescence microscope (Zeiss Axio Observer), an A1R; C2 (Nikon) confocal microscope; Eclipse Ti (Nikon) spinning disk confocal microscope	NR	[1, 2, 3 d]*, 4, 5, 6 d	Yolk sac, tail, ICM, PBI and AGM	**+**
Hamilton et al. [21]	FITC	Spinning Disk confocal microscope	NR	1, 4, 7, 10, 13 h	Tail	**+**
Staal et al. [24]	PKH26	Leica fluorescent microscope	NR	NR	Yolk sac, tail	**+**
Lopez et al. [22]	GFP	Leica MZ16FA microscope, using a Hamamatsu digital camera model C7780 and Volocity Imaging software (Perkin Elmer)	NR	NR	NR	**+**
Ushiki et al. [25]	Cy5.5; AF750	IVIS Spectrum system (Xenogen, Alameda, CA, USA)	Excitation/emission (nn): 640/700 for Cy5.5, 710/780 for AF750. Exposure time: 5 s, lamp level: high, binning: medium, FOV: 12.9612.9 cm, and f/stop: 1.	0.5, 1, 3, 6, 12, 18, 24 h	Bone marrow, lung, spleen, liver, kidney	**+**

**Abbreviation**s—Ref.: reference; CFSE: 5-(6)-carboxyfluorescein diacetate succinimidyl Ester; FITC: Fluorescein isothiocyanate; GFP: green fluorescent protein; Cy5.5: cyanine 5.5; AF750: Alexa Fluor 750; ICM: intermediate cell mass; PBI: posterior blood island; AGM: aorta-gonad-mesonephros; NR: not reported; s: seconds; cm: centimeter; d: day; h: hours; min: minutes; nn: nanometer, FOV: field of view. **Note**—* means the FLI intensity signal detectable.

**Table 10 cells-09-00939-t010:** HSC migration homing and tracking by magnetic resonance imaging.

Ref.	Equipment	Software	MF (T)	Sequence	Weighted Images (TR/TE; ms)	FOV;MT;ST	Homing Evaluation Time	Uptake Distribution	Outcome (Cells Migration)
Sweeney et al. [34]	Varian^®^ Unity/INOVA 4.7 T small animal scanner	MIPAV	4.7	Fast Spin Echo; Gradient Echo	T2: 2100/15 T2*: NR	NR; 256x512; NR	3, 24 h (6, 9 d)	Bone marrow, spleen, and liver	+
Bengtsson et al. [32]	Bruker BioSpin, Madison, WI, USA	Paravision 4.0 (Bruker, Madison, WI, USA) and OsiriX v.3.5	17.6	3D Gradient Echo	T2*: 80/2.5	1.1x0.6x0.5cm^3^; 393x214x83; NR	7, 14 d	Bone marrow	+
Niemeyer et al. [33]	1.5-T imaging MR scanner (ACS NT; Philips, Best, the Netherlands)	NR	1.5	3D Fast Field Echo	T2*: 32/14	100x80mm; 512x512; 0.4mm	2, 24 h	Bone marrow and liver	+

**Abbreviations**—Ref.: reference; 3D: three dimensional; MF: magnetic field; TR: repetition time; TE: echo time; T2: transversal relaxation time; T2*: transversal relaxation time star; FOV: field of view; MT: matrix; ST: Slice thickness; NR: not reported; T: tesla; ms: millisecond; mm: millimeter; d: day; h: hours.

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
