# Peer review of "Noninvasive Tracking of Hematopoietic Stem Cells in a Bone Marrow Transplant Model"

_cells, 2020, doi:10.3390/cells9040939_

Round 1
Reviewer 1 Report
The authors present a very thorough and detailed review of the use of noninvasive techniques for tracking stem cells following bone marrow transplantation. The review strategy is clearly stated and the exclusion and inclusion criteria clearly state the objective and function of the review article. I have no concerns about the choice of papers used to formulate the review.
The authors have included detailed figures and tables for all the key points of the review including animal models, source of label, detection method and summarize the outcomes of these studies. Again, I cannot fault the thoroughness of the authors.
I have no suggested changes and feel that this is an excellent review to someone new to the field as it summarizes the history and current state of these HSC tracking studies highlighting both the utility and limitations of the methods.
Author Response
Reviewer #1
- The authors present a very thorough and detailed review of the use of noninvasive techniques for tracking stem cells following bone marrow transplantation. The review strategy is clearly stated and the exclusion and inclusion criteria clearly state the objective and function of the review article. I have no concerns about the choice of papers used to formulate the review.
The authors have included detailed figures and tables for all the key points of the review including animal models, source of label, detection method and summarize the outcomes of these studies. Again, I cannot fault the thoroughness of the authors.
I have no suggested changes and feel that this is an excellent review to someone new to the field as it summarizes the history and current state of these HSC tracking studies highlighting both the utility and limitations of the methods.
Answer: Thank you for your time and dedication in the manuscript review, as also for the recognition of our effort in preparing this manuscript.
Reviewer 2 Report
In the introduction, The authors have described individual molecular imaging modalities mostly regarding the targets(disease) and techniques. This section is not well-structured, might bring difficulties to follow. Also, It could help to re-structure or category into two-three paragraphs following the importance of in vivo tracking to disease modeling (graft) to techniques.
Authors have summarized the extraction and isolation of HSC, classified the species, techniques, and outcome. Subsequently, the authors described the ex vivo or in vitro labeling strategies. Other than MRI or other anatomical imaging modalities, the radionuclide labeling for PET or SPECT nuclear imaging is not a sort of contrast agent. The authors could use the term “labeling strategies or techniques” instead of current content.
The authors mainly summarized the application of each individual imaging-tech; But much less on the imaging self in terms of acquisition or quantification approach.
The authors have discussed each imaging modality individually, and I highly suggest to add A clear illustration or table to summarize all the categorized imaging techniques apart from agents sorts (such as optical imaging, nuclear imaging, magnetic imaging) could help to understand the advantages and disadvantages of each imaging tech. The comparison could comprise tracers/contrast agents (bio half-life, radio half-life); application limitation, particular in clinical potential; physical features of imaging (acquisition, reconstruction, quantification, and so on).
The authors also need to discuss more clinical or non-preclinical academical potentials of stem cell tracking, including diseases (oncological, non-oncological..), also in comparison/combination to the non-imaging approach.
Author Response
March 31, 2020
Cells-738051
Noninvasive tracking of hematopoietic stem cells in a bone marrow transplant model
To
Assigned Editor: Marija Dragojevic
Cells
Dear Editor,
We are sending the revised version of the manuscript entitled “Noninvasive tracking of hematopoietic stem cells in a bone marrow transplant model”, Manuscript cells-738051, with point-by-point corrections (see below) suggested by reviewer 2. The changes in the manuscript have been highlighted using the “Track Changes” option.
Thank you again for your time and consideration. We hope the paper is now suitable for publication in Cells. We look forward to hearing your decision.
Sincerely,
Lionel Gamarra
Reviewer #2
- In the introduction, The authors have described individual molecular imaging modalities mostly regarding the targets (disease) and techniques. This section is not well-structured, might bring difficulties to follow. Also, It could help to re-structure or category into two-three paragraphs following the importance of in vivo tracking to disease modeling (graft) to techniques.
Answer: Thank you for your consideration. We modified this part of the introduction section of the manuscript, restructuring the text in the following molecular imaging categories: optical, nuclear and MRI. We also described the strengths and weaknesses of each imaging modalities regarding their technical peculiarities, engraftment evaluation, translational stage, and suitability to monitor HSC transplantation.
- Authors have summarized the extraction and isolation of HSC, classified the species, techniques, and outcome. Subsequently, the authors described the ex vivo or in vitro labeling strategies. Other than MRI or other anatomical imaging modalities, the radionuclide labeling for PET or SPECT nuclear imaging is not a sort of contrast agent. The authors could use the term “labeling strategies or techniques” instead of current content.
Answer: Thank you for your suggestion. We agree and modified the 3.4 item title for "Labeling strategies and techniques of hematopoietic stem cells with tracers", as well as the term "contrast agent" for tracers, adopted in different molecular imaging studies [1-3].
- The authors mainly summarized the application of each individual imaging-tech; But much less on the imaging self in terms of acquisition or quantification approach.
Answer: Thank you for your consideration. In terms of the acquisition, the major parameters of imaging acquisition for each technique were described in tables 7 to 10. Overall, the studies on optical imaging (bioluminescence and fluorescence, Tables 7 and 9, respectively) reported few imaging acquisition parameters. The probable reason for this is that optical imaging hardwares and softwares provide in-built correction algorithms which convert raw counts into calibrated units of bioluminescence or fluorescence. Thus, the measured signal intensity remains practically unaltered regardless of the acquisition parameters such as exposition time, camera opening, distance from detector and binning. On the other hand, MRI studies reported imaging acquisition parameters in more details (Table 10). We nevertheless complemented the results section of the manuscript with additional information about the imaging acquisition parameters described in the Tables. Regarding the quantification approaches, the selected studies used imaging data only to demonstrate graft expansion rather than to quantify chimerism. To access the degree o chimerism in a quantitative manner, the selected studies relied on techniques other than imaging described in the items “Graft assessment” and "Graft efficiency" of Table 6, respectively. Furthermore, the intention of this systematic review was to highlight the strengths and weaknesses of each imaging technology to track cells in real-time. The impact of quantification approach in this specific context is far less important than other parameters such the principle of the technology, penetration of the signal through tissues, stability of the reporter probes/molecules, etc. This is the reason why those parameters were described in more details in this review.
- The authors have discussed each imaging modality individually, and I highly suggest to add A clear illustration or table to summarize all the categorized imaging techniques apart from agents sorts (such as optical imaging, nuclear imaging, magnetic imaging) could help to understand the advantages and disadvantages of each imaging tech. The comparison could comprise tracers/contrast agents (bio half-life, radio half-life); application limitation, particular in clinical potential; physical features of imaging (acquisition, reconstruction, quantification, and so on).
Answer: Thank you for your consideration. We elaborate the Figure 3 in according to the reviewer request and added two paragraphs in the discussion section of the manuscript, in which was included the considerations requested by the reviewer on molecular imaging modalities.
- The authors also need to discuss more clinical or non-preclinical academical potentials of stem cell tracking, including diseases (oncological, non-oncological..), also in comparison/combination to the non-imaging approach.
Answer: Thank you for your consideration. We added complementary information about theses aspects in the discussion section of the manuscript.
References
- Chen, K.; Chen, X. Design and development of molecular imaging probes. Curr Top Med Chem 2010, 10, 1227-1236, doi:10.2174/156802610791384225.
- Perrone-Filardi, P.; Dellegrottaglie, S.; Rudd, J.; Costanzo, P.; Marciano, C.; Vassallo, E.; Marsico, F.; Ruggiero, D.; Petretta, M.; Chiariello, M., et al. Molecular imaging of atherosclerosis in translational medicine. European journal of nuclear medicine and molecular imaging 2010, 38, 969-975, doi:10.1007/s00259-010-1697-5.
- Welling, M.M.; Hensbergen, A.W.; Bunschoten, A.; Velders, A.H.; Roestenberg, M.; van Leeuwen, F.W.B. An update on radiotracer development for molecular imaging of bacterial infections. Clinical and Translational Imaging 2019, 7, 105-124, doi:10.1007/s40336-019-00317-
Round 2
Reviewer 2 Report
The author has adequately revised the manuscript. it's acceptable for publication.